# The Great Escape: The Power of Cancer Stem Cells to Evade Programmed Cell Death

**DOI:** 10.3390/cancers13020328

**Published:** 2021-01-17

**Authors:** Vanessa Castelli, Antonio Giordano, Elisabetta Benedetti, Francesco Giansanti, Massimiliano Quintiliani, Annamaria Cimini, Michele d’Angelo

**Affiliations:** 1Department of Life, Health and Environmental Sciences, University of L’Aquila, 67100 L’Aquila, Italy; vanessa.castelli@univaq.it (V.C.); elisabetta.benedetti@univaq.it (E.B.); francesco.giansanti@univaq.it (F.G.); mquintiliani@unite.it (M.Q.); 2Department of Medical Biotechnology, University of Siena, 53100 Siena, Italy; antonio.giordano@temple.edu; 3Sbarro Institute for Cancer Research and Molecular Medicine and Center for Biotechnology, Temple University, Philadelphia, PA 19122, USA

**Keywords:** cancer stem cells, programmed death, therapeutic approaches, recurrence, metastasis

## Abstract

**Simple Summary:**

Several current therapies fail to eradicate tumors due to the ability of cancer stem cells (CSCs) to escape different programmed cell deaths. In fact, apoptosis, autophagy, and necroptosis are deregulated in CSCs. Over the years, it has emerged that CSCs play a key role in tumor formation, progression, and recurrence. Thus, developing CSC-selective and programmed death-inducing therapeutic approaches appears of primary importance. In this review, we focus on signaling pathways involved in CSCs’ escape-death ability and the variety of CSC-targeting agents developed in recent years.

**Abstract:**

Cancer is one of the primary causes of death worldwide. Tumour malignancy is related to tumor heterogeneity, which has been suggested to be due to a small subpopulation of tumor cells named cancer stem cells (CSCs). CSCs exert a key role in metastasis development, tumor recurrence, and also epithelial–mesenchymal transition, apoptotic resistance, self-renewal, tumorigenesis, differentiation, and drug resistance. Several current therapies fail to eradicate tumors due to the ability of CSCs to escape different programmed cell deaths. Thus, developing CSC-selective and programmed death-inducing therapeutic approaches appears to be of primary importance. In this review, we discuss the main programmed cell death occurring in cancer and the promising CSC-targeting agents developed in recent years. Even if the reported studies are encouraging, further investigations are necessary to establish a combination of agents able to eradicate CSCs or inhibit their growth and proliferation.

## 1. Introduction

Cancer is one of the primary causes of death worldwide [1]. Tumor malignancy is related to tumor heterogeneity, which has been suggested to be due to a small subpopulation of tumor cells named cancer stem cells (CSCs) [2]. The first isolation of CSCs by fluorescence-activated cell sorting using CD34 and CD38 (CD34+CD38−) surface marker expression was in acute myeloid leukemia and dates back to 1994 [3]. Since then, CSCs have been isolated from numerous solid tumors, including breast, colon, lung, brain, and prostate tumors using FACS and magnetic cell sorting [4,5,6,7,8,9]. To date, CSC recognition has been mostly based on surface markers as well as their capability to propagate and self-renew. Nevertheless, CSC surface markers alone are not a consistent method of detecting these subpopulations, thus leading to some uncertainty and disagreement in the field. This is due to the fact that it is improbable to use a universal specific marker for the detection of these cells. A few functional markers such as the ATP-binding cassette (ABC) transporter and aldehyde dehydrogenase (ALDH), live-cell RNA, single-cell DNA detection, and the activation of some crucial signaling pathways have made identification of CSCs easier [10].

To date, numerous methods have been proposed to detect, characterize, and isolate CSCs (also known as tumor-initiating cells). The approaches are marker-based or marker-independent. Both approaches involve the use of FACS and magnetic-activated cell sorting. The marker-based approaches include the antigenic approach for numerous cell surface markers, such as epithelial-specific antigens CD133, CD44, CD24, and CD166. However, this method shows numerous disadvantages since these markers are not solely expressed by CSCs [11]. An example of the marker-independent strategy is represented by side population (SP), which was originally used to enrich hematopoietic stem cells and leukemia stem cells but was afterward widely utilized to enrich CSCs in different solid tumors [12,13]. This approach is based on the capability of stem cells to exclude Hoechst dye 33342 and appear as a side population compared with normal differentiated cells, which take up the dye and appear as the major population. ABC transporter family proteins are responsible for dye exclusion and the side population phenotypic profile. Additionally, in this case, there is a problem with overlaps [14]. CSCs have been detected in different types of tumor, and specific markers have been used to isolate and characterize them (Table 1).

Like normal pluripotent stem cells, CSCs display the properties of asymmetric division, the quiescent potentials in a dormant state, a role in the epithelial–mesenchymal transition (EMT), apoptotic resistance, self-renewal, tumorigenesis, differentiation, and drug resistance [41]. Furthermore, CSCs exert a key role in metastasis development and in tumor recurrence [42]. CSCs often show the re-expression of embryonic markers comprising Oct4, Sox2, and Nanog and a distinct metabolic profile from terminally differentiated tumor cells; they inhabit specialized hypoxic microenvironments that provide long-term maintenance [43]. CSC target therapies have long been suggested in conjunction with traditional chemotherapeutic procedures to eradicate both differentiated cells and CSCs and avoid forms of recurrence [44]. 

CSCs show self-renewal and flexible clonogenic properties and help identify precise tumor microenvironments [45]. The communication between CSCs and their tumor niche is firmly related to the characterization of CSCs [46]; based on this, CSCs can maintain the tumor heterogeneity that causes the malignant behaviors of metastasis, invasion, and drug resistance [47]. The impact of the tumor microenvironment on CSC physiology has been demonstrated to be via intrinsic and extrinsic mechanisms. The intrinsic mechanisms comprise DNA demethylation or methylation and gene mutation, while the extrinsic pathway includes the release of numerous growth factors and cytokines by the tumor microenvironment, inducing the initiation of specific signaling pathways [46]. Moreover, different studies have reported that CSCs may be responsible for tumor resistance to traditional therapy (chemo- and radio-resistance). This resistance is increased by the crosstalk between CSCs and the tumor microenvironment, which is characterized by enhanced resistance to hypoxia, the activation of the DNA repair system, and the EMT [48,49]. These mechanisms are at the basis of the therapeutic failures often faced in various tumor situations. Thus, both CSCs and the tumor microenvironment represent relevant therapeutic targets. Several current therapies fail to eradicate tumors due to the ability of CSCs to escape different programmed cell deaths (Figure 1); the residual CSCs can survive and promote cancer recurrence. In this review, we will focus on the main programmed cell deaths and the capacity of CSCs to elude them. Moreover, we discuss a variety of CSC-targeting agents developed in recent years. The exposed evidence highlights that a combination of modulators of programmed cell death and standard chemotherapeutic agents may be an efficient anticancer therapeutic approach. This review aims to encourage researchers to develop and investigate approaches based on the modulation of the reported mechanisms.

## 2. Apoptosis

Apoptosis is a regulated programmed cell death that plays a key role in tissue development and homeostasis [50]. Apoptosis is, therefore, a highly controlled and regulated process that is necessary for proper functioning and development, not only for tissues but for the whole organism. In fact, the correct balance between survival and death signals is a crucial point—a failure in regulation can lead to the development of tumors [51]. In this context, apoptosis plays an important role in preventing cancer development, such as inducing cells to self-destruct after radiation-induced DNA damage or anticancer drugs [52]. A tumor characteristic linked to specific genetic mutations is precisely the ability to evade the death process and not respond to drugs that induce apoptosis. Many studies have shown that genetic mutations transform normal stem cells into CSCs and enable them to escape apoptosis, which leads to tumor formation (extensively reviewed in [51]). Another way to create apoptosis-resistant CSCs involves dedifferentiation and reprogramming. In particular, numerous lines of evidence have established that CSCs have the potential to transdifferentiate into other lineage cells (pericytes vascular and endothelial cells) for promoting tumor growth and metastasis in some tissue contexts instead of only recruiting stromal cells from neighboring or distant tissues [53]. 

CSCs inhabit niches that are crucial for their stemness [54]. Non-CSCs in the tumor microenvironment may shield CSCs from chemotherapy [55]. Since CSCs can transdifferentiate into stromal cells in the tumor microenvironment, selectively targeting these CSC-derived cells may disturb the CSC niches and may represent a potential approach for cancer therapy [56]. Nevertheless, to date, there are no specific drugs targeting CSCs. Therefore, inducing the differentiation of CSCs and targeting the CSC-derived stromal cells may help in developing new diagnostic and therapeutic strategies against tumors and may have a clinical advantage when treating cancer.

In particular, CSCs show overexpression of antiapoptotic proteins, including phosphatidylinositol 3-kinase (PI3K)/Akt, NOTCH1, and Wnt/β-catenin, and fast repair of the DNA that led to apoptosis resistance [51]. These characteristics of CSCs make them among the main failure factors of anticancer therapies. In fact, the ability of CSCs to evade programmed cell death leads to resistance to various chemotherapeutic agents, such as those used in leukemia, malignant melanoma, brain, head and neck cancers, and breast, colorectal, and pancreas tumors [51,57,58,59]. Furthermore, resistance to radiotherapy in brain and breast cancers has also been demonstrated. Thus, traditional anticancer treatments can reduce or eliminate cancer cells, but CSCs can survive, leading to the formation of relapses in many types of cancer or metastases by migrating from the primary site of the tumor [59]. In fact, the relative abundance of CSCs is associated with the clinical outcome [59]. We can distinguish two main pathways of apoptosis: intrinsic and extrinsic [60].

### 2.1. Intrinsic Pathway

The intrinsic pathway of apoptosis is based on mitochondria and mitochondrial proteins. DNA damage, oncogenes, growth factor deprivation, microtubule-targeting drugs, DNA-damaging molecules, and Ca^2+^ surplus can induce this programmed death pathway [61]. P53, following damage to DNA, can induce the intrinsic pathway thanks to the activation of PUMA (P53-upregulated modulator of apoptosis) and Bcl2-associated X protein (BAX), which induce mitochondrial outer membrane permeabilization (MOMP) [62]. MOMP induces the release of cytochrome c, the second mitochondria-derived activator of caspase (SMAC) and Omi; this is considered the point of no return of the intrinsic pathway. After their escape from the mitochondrion and their arrival in the cytoplasm, the cytochromes bind to apoptotic protease-activating factor-1, dATP, and procaspase-9, forming the apoptosome. Then, the apoptosome causes the monomers of procaspase-9 to unite into dimers and form the active protein (caspase-9), which triggers the executioner caspases-3 and -7. The executioner caspases rapidly act on target proteins, including nuclear lamina, ICAD/DFF45, poly ADP ribose polymerase (PARP), and P21-activated kinase, inducing cell death [63].

Cell death triggered by the intrinsic pathway is also enhanced by other mechanisms. Omi (also known as serin protease HtrA2) inhibits caspase inhibitor X-linked inhibitor of apoptosis protein (XIAP), and mitochondria-derived activator of caspase (MAC) is released from the mitochondrion, blocking inhibitor of apoptosis proteins. Furthermore, MOMP can induce cell death in an independent caspase process [64,65,66,67]. However, in some tumors, such as breast [68,69,70], ovary, and skin cancers, it has been shown that CSCs can evade death upon mitochondrial dysfunction [57]. Numerous investigations in colon or breast cancer cells have highlighted a correlation between increased aggressiveness and nonlethal caspase activation [71,72,73]. In a recent interesting study, Berthenet [74] and collaborators in melanoma cells demonstrated that the resilience of cancer cells to experience complete apoptosis conducts to a more aggressive cancer phenotype. Furthermore, they provide a model to induce and examine the effects of failed apoptosis in physiological settings, such as neuronal function or stemness [74]. Failed apoptosis also seems to influence the emergence of CSCs in breast cancer. In particular, it has been observed that breast cancer cells surviving staurosporine gained metastatic potential in vivo, while some of them developed CSC properties [72].

This point is of high relevance to the scientific community because dissecting the underlying mechanisms of failed apoptosis-driven transcriptional signature (which increases the aggressiveness of a tumor) may improve the targeting of metastatic processes.

### 2.2. Extrinsic Pathway

Extracellular signals can induce apoptosis via the extrinsic pathway. The main cell death signals are Fas ligand (Fas-L), TNF-related apoptosis-inducing ligand (TRAIL), and tumor necrosis factor (TNF), also called death ligands [75]. These signals bind to tumor necrosis factor (TNF) family death receptors, inducing adaptor protein recruitment that includes the Fas-associated death domain (FADD) and the TNF-receptor-associated death domain (TRADD) [75]. The adaptor protein binds initiator procaspases-8 and -10, forming the death-inducing signaling complex (DISC) and converting procaspases-8 and -10 to their active form, which further activates effector caspases-3, -6, and -7 [76]. These effector caspases lead to cell death by the cleavage of proteins and the cytoskeleton. The extrinsic and intrinsic pathways meet after the activation of caspase-8. DISC is a multiprotein complex composed of members of the death receptor family of apoptosis-inducing cellular receptors [77,78,79]. TNF binds to TNFR1 and TNFR2. However, only TNFR1 contains a death domain and is part of the death receptor subfamily [80,81]; the interaction between TNF and TNFR1 leads to the formation of the TNFR1 signaling complex (TNFR1-SC). 

In the extrinsic pathway, the activation of caspase-8 induces BH3 interacting-domain death agonist (BID) activation, a BH3-only protein [60]. BID, in turn, oligomerizes BAX and BCL2 homologous antagonist/killer (BAK), and the intrinsic apoptotic pathway proceeds. Both pathways will keep on propagating via their typical course, supporting apoptosis (Figure 1A) [60].

### 2.3. Mechanisms of CSCs to Evade Apoptosis

Apoptosis, a natural mechanism of cell death, can be used as an effective tool in anticancer therapy [61,82,83]. In fact, there are several anticancer drugs that target apoptosis in both intrinsic and extrinsic ways. This strategy represents the most effective nonsurgical treatments [51]. However, CSCs show an innate resistance to apoptosis through several mechanisms, including multidrug resistance transporters. ABC family overexpression in CSCs is involved in multidrug resistance. The ABC family is characterized by 49 members, including multidrug resistance protein 1 (MRP1, ABCC1), breast cancer resistance protein (BCRP, ABCG2) [84,85,86,87], MRP5/ABCC5 [87], and P-glycoprotein (P-gp, MDR1, ABCB1) [88]. It has been shown that CSCs are upregulated in numerous cancers, including colon, glioblastoma, prostate, and pancreas, contributing to drug resistance [89,90,91,92,93]. In addition to the ability to transport the drug outside the cell, reducing the amount needed to trigger cell death, it has been proposed that these transporters contribute to tumorigenesis [94,95]. Given the role of ABC in drug resistance, several therapies have been tried over the years; however, the targeting of ABC transporters in cancer patients has been unsuccessful [84]. 

Another pathway involved in the evading mechanism from CSCs is PI3K/AKT/mTOR signaling. This pathway is fundamental for proliferation, metabolism, invasion, and survival and, therefore, for tumor development and the maintenance of CSCs [96]. PI3K/AKT/mTOR inhibitors have been developed and tested in recent years, including salinomycin, metformin, silibinin E1201, rottlerin, and torin [97]. Yang and collaborators demonstrated that resistance to TMZ can be increased in GBM by using metformin; metformin in combination with sorafenib as an RAF inhibitor also substantially diminished CSC oxidative stress and efflux pump activity and eradicated these tumor cells [98]. Moreover, another inhibitor of the PI3K/AKT/mTOR pathway, BFZ-235, induced a stemness reduction of colon CSCs [99].

Furthermore, an alteration of the ratio between apoptotic and antiapoptotic proteins is essential in the development of various tumors and contributes to the maintenance of CSCs. However, their contribution to drug resistance has not yet been well characterized. In CSCs, a high expression in BCL2 family proteins, which consist of proapoptotic proteins Bax, Bak, Bid, Bim, Bik, Noxa, and Puma and antiapoptotic molecules Bcl-2, Bcl-XL, and Mcl-1, was observed [100,101]. The altered relationship between pro- and anti-apoptotic proteins is related to the resistance of CSCs to apoptosis and anticancer therapies. In addition, in CSCs, a large increase in the expression of nuclear factor erythroid 2-related factor 2 (Nrf2) can be observed [102]. Nrf2 is the redox-sensing transcription factor that stimulates the expression of ABCG2, Bcl-2, and Bmi-1, a member of polycomb repressor complex (PRC1) genes, promoting CSC survival [103]. Based on this evidence, approaches targeting these antiapoptotic proteins are relevant in counteracting the resistance to apoptosis from CSCs.

TRADD is involved in several receptor signaling pathways and has a key role in different biological processes, including cell survival and apoptosis in different cellular contexts [104]. Extracellular TNF can activate TNFR1, which, in turn, leads intracellular proteins to activate numerous signaling pathways, such as mitogen-activated protein (MAP) kinase pathways and nuclear factor κB (NF-κB) [105]. In CSCs, TRADD has an essential role in NF-κB activation, in which the exposed nuclear localization signal enters the nucleus and binds to specific sequences of DNA for prosurvival signaling [106,107]. NF-κB is the transcription factor that stimulates the expression of a variety of inflammatory cytokines and apoptosis inhibitory proteins [108,109]. A high expression of TRADD, enough to activate NF-κB, was observed in glioblastoma [110]. Furthermore, a reduction in cell viability was observed by reducing the activation of NF-κB through the silencing of TRADD, confirming the role of TRADD in the survival of CSCs [110]. The use of NF-κB inhibitors, such as parthenolide, pyrrolidinedithiocarbamate, and diethyldithiocarbamate, showed an induction of apoptosis in human prostate and breast cancer CSCs but not in normal stem cells, indicating a potential use of this inhibitor in cancer therapy [111,112]. 

The inhibitor of apoptosis (IAP) family proteins directly or indirectly inhibit the apoptotic cascade and play an essential role in the survival of tumor cells, especially in CSCs [113]. Furthermore, some IAPs are involved in the activation of the NF-κB pathway and positively regulate cell survival. The IAP family proteins are composed of numerous proteins involving cIAP2, survivin, IAP1, ML-IAP, XIAP, NAIP, and ILP-2 [114]. IAPs constrain the activity of caspases-3, -7, and -9, implicated in the evasion of tumor cells from apoptotic death [115]. In human cancers, the expression or function of IAP proteins is deregulated, and the expression levels of IAP proteins and their antagonists have been associated with clinical parameters and cancer prognosis [116,117]. Among the IAP family members, XIAP may have the strongest antiapoptotic activities compared to other IAPs, upon a specific stimulus (i.e., depletion of cIAP1/2) [118,119,120,121]; it prevents caspase-9 activation, binding to active caspases-3 and -7 [122]. It has been reported that XIAP has the function of directly inhibiting caspase, and structural and biochemical studies have accurately mapped the elements of XIAP that are necessary for caspase inhibition, which, interestingly, are not conserved among IAPs [123,124]. Indeed, it has been shown that XIAP is probably the only real caspase inhibitor, implying that the other family members have never achieved the capability to prevent caspase activity directly [123,125].

XIAP proteins play a key role in regulating apoptosis in CSCs and are expressed at higher levels in glioblastoma and nasopharyngeal carcinoma stem cells [126]. IAP inhibitors have been successfully tested in glioblastoma, ovarian cancer, and colon and nasopharyngeal carcinoma [126]. In a radioresistant glioblastoma cell line, XIAP inhibitors favored apoptosis induced by gamma-irradiation [127]. 

FLICE-inhibitory protein (c-FLIP) is the main antiapoptotic protein that cooperates with FADD, caspases-8 or -10, and DR5 to inhibit the formation of DISC and the resultant activation of the caspase cascade. Responsible for resistance to chemotherapy-induced apoptosis, C-FLIP is overexpressed in several cancers, such as breast cancer, glioblastoma, and leukemia; however, its levels in CSCs are much higher than in normal cancer cells [128,129], inducing resistance to TRAIL-induced apoptosis. In fact, cFLIP silencing makes CSCs sensitive to TRAIL-induced apoptosis, indicating the role of cFLIP in death resistance [128,129]. 

The use of specific drugs against c-FLIP isoforms can improve the effectiveness of chemotherapy treatments. Sorafenib, a broad-spectrum kinase inhibitor targeting the RAF–MEK–ERK pathway, enhanced sensitization to TRAIL- or FAS-mediated apoptosis by downregulation of cFLIP in endometrial carcinoma cell lines that are resistant to TRAIL or FAS [130]. Piggott et al. [131] showed that the use of a cFLIP inhibitor, in combination with TRAIL treatment, caused a reduction in tumor growth and metastasis formation in an in-vivo model of breast cancer. The treatment inhibited CSC self-renewal but did not induce the death of healthy mammary cells [131]. 

It has been demonstrated that CSCs are resistant to DNA-damaging therapies by controlling the cell cycle, improving DNA repair capacity and the scavenging of reactive oxygen species [132,133,134,135]. Interestingly, Bartucci et al. [136] reported that the co-administration of Chk1 inhibitor AZD7762 and chemotherapy inhibited nonsmall-cell lung (NSCL) CSC growth in mouse xenografts in a p53-independent manner. Additionally, targeting CHK1 and PARP1 may be an efficient anti-CSC approach [132].

## 3. Ferroptosis

CSCs can be eliminated by ferroptosis, which is a nonapoptotic-regulated mechanism of cell death [137]. Ferroptosis is a nonapoptotic, regulated cell death characterized by abnormal metabolism of cellular lipid oxides catalyzed by iron ions or iron-containing enzymes. During ferroptosis, numerous inducers eliminate the cell redox balance and generate a large number of lipid peroxidation products, ultimately causing cell death [138]. In contrast to normal cancer cells, in CSCs, enhanced iron content has been found. Consistently, iron chelation may reduce their stemness, while iron supplements may counteract this effect, exerting protective effects in CSCs [139,140]. Accordingly, iron deficiency impeded cell proliferation in mouse-induced pluripotent cells and inhibited the expression of stemness markers [141]. These results suggest that iron may exert a dual function in CSCs. Different investigations have reported that the levels of the iron exporter ferroportin as well as storage protein ferritin are reduced in ovarian CSCs and cholangiocarcinoma CSCs [139,142]. Moreover, reduced expression of ferritin and high transferrin receptor levels were found in cholangiocarcinoma cells growing in monolayers, while higher levels of ferritin and low expression of transferrin receptor were reported in the same cell lines growing in spheroids [142], suggesting that iron is crucial for spheroid formation. The knock-down of ferritin counteracted glioblastoma CSC growth and reduced a stem-like phenotype in breast cancer [143,144]. These investigations demonstrate that CSCs are iron-rich and iron-dependent. Notably, ovarian CSCs showed greater sensitivity to ferroptosis than nontumorigenic ovarian stem cells [142]. As mentioned above, ferritin exerts a dual role in CSCs: in slow-growing CSCs, iron is not essential for proliferation and may be stored as ferritin, thus avoiding the generation of lipid peroxidation and explaining the resistance of cholangiocarcinoma cells with high ferritin content to erastin. Similarly, if ferritin degradation starts, ferritin may also represent a source of iron, thus causing ferroptosis [141]. Recently, Müller et al. reported a fascinating role for iron in inducing the expression of CSC marker CD44 [145]. They also show a new mechanism of iron entry in cells under the mesenchymal state. Similar to transferrin receptor 1, CD44 may also perform as an alternative route for iron entry when the transferrin receptor is downregulated due to higher intracellular levels of iron [145], confirming that CSCs are dependent on high intracellular iron levels. Indeed, iron chelation was linked to the downregulation of stemness genes, CSC surface markers including CD133, CD44, and CD24, as well as EMT-inducing transcription factors [141]. Overall, the induction of ferroptosis is an attractive strategy to eradicate tumors due to its ability to selectively target aggressive CSCs.

## 4. Necroptosis

Necroptosis exerts a crucial role in the regulation of cancer biology, comprising cancer subtypes, cancer immunity, cancer metastasis, and oncogenesis [146,147]. Necroptosis is a type of regulated necrosis mediated by death receptors [148,149]. The key mediators of the necroptotic pathway, alone or combined with other programmed cell deaths, have been indicated to encourage cancer progression and metastasis [150,151,152]. Contrarily, necroptosis also apparently operates as a “fail-safe”, protecting against tumor progression when apoptosis is altered [153,154]. Based on its pivotal role in cancer biology, necroptosis has appeared as a new target for cancer treatment; an increasing number of therapeutic agents have shown potential in counteracting tumor growth and recurrence by stimulating or controlling necroptosis [155]. Necroptosis is determined by both the induction of mitochondrial uncoupling proteins and the blocking of oxidative phosphorylation in CSCs [156]. Necroptosis is also related to the ALDH family, which may represent a potential therapeutic target for CSCs (Figure 1B) [157].

## 5. Autophagy

Autophagy is a highly conserved catabolic process involved in organelle turnover, protein degradation, and nonselective breakdown of cytoplasmic components. There are three primary kinds of autophagy in mammalian cells: microautophagy, chaperone-mediated autophagy (CMA), and macroautophagy [158]. All types of autophagy finish in the delivery of cargo to the lysosome for degradation and recycling, but they are morphologically different. Microautophagy uses lysosomal membrane invaginations to capture cargo. Instead, during CMA, chaperones are used to identify cargo proteins without using membranous structure; the proteins with a pentapeptide motif are unfolded and translocated directly across the lysosomal membrane [159]. 

In contrast to microautophagy and CMA, macroautophagy includes the elimination of cargo away from the lysosome. It is based on the formation of autophagosomes and double-membrane vesicles and happens at the basal level in all cells. It can be caused by several signals and/or cellular stressors, including hypoxia, radiation, deficiencies in nutrients and energy, damage of mitochondrial/DNA, and endoplasmic reticulum (ER) stress [160,161]. The autophagic process can be divided into five steps: initiation, elongation and autophagosome formation, fusion, and autolysosome formation and degradation. Autophagosome production and autophagy are regulated by over 20 core autophagy genes (ATGs) that are highly conserved [160]. 

Autophagosome formation (initiation) is required in the mammalian cell Unc51-like kinase 1 (ULK1) complex formed by ULK, Atg13, Atg101, and FIP200 [162,163,164,165]. The target of rapamycin (mTOR) complex 1 is the main negative regulator of autophagy, inhibiting the activation of the ULK1 complex. Autophagosome elongation and maturation require two ubiquitin-like conjugation systems, including the microtubule-associated protein 1light chain 3 (LC3) system and the Atg12 system (associated with Atg5 by Atg10 and Atg7) [166]. Autophagic membrane elongation is promoted by the Atg12–Atg5 complex interacting with Atg16L. Concomitantly, the LC3 (LC3-I) cytosolic form is conjugated to phosphatidylethanolamine by Atg7 and Atg3, forming the insoluble form LC3-phosphatidylethanolamine conjugate (LC3-II), which is inserted into the autophagosomal membrane. Finally, the autophagosome, including organelles and cargo proteins, is fused with a lysosome to form an autolysosome, which degrades its content into fatty acids, amino acids, and nucleotides (Figure 1C) [167].

Autophagy is a necessary component to maintain pluripotency in response to various stimuli. Pluripotency is an important characteristic of CSCs, namely, to self-renew and maintain the undifferentiated state [168]. In fact, it has been observed that in several CSCs such as in breast [169,170], pancreatic, liver [171], osteosarcoma [172], ovarian [173], and glioblastoma [174], a high autophagic flow is maintained. Furthermore, it has been proposed that autophagy and hypoxia are essential for the maintenance of the stem cell niche. Indeed, Zhu et al. demonstrated that autophagy is HIF-1α-dependent, and it is important for keeping the equilibrium between pancreatic CSCs and normal cancer cells [175]. Thus, autophagy is an adaptive mechanism necessary for CSC maintenance. The cargo of the autophagosome, including organelles and proteins, is finally degraded, fusing with a lysosome and generating the autophagolysosome. In this step, clinically available therapeutic approaches, comprising chloroquine (CQ) and hydroxychloroquine, impede autophagy, reducing autophagosome and lysosome fusion [160]. Other drugs are in development to target this stage, including D1661, which is an inhibitor of both mTOR and autophagy.

### 5.1. Mechanisms of CSCs to Evade Autophagy

One of the crucial mechanisms that have been firmly correlated with CSC survival, pluripotency, and aggressiveness is autophagy. In fact, autophagy and autophagy proteins are upregulated in breast CSCs [169,176] with respect to adherent cells. Successively, autophagy has been highlighted in numerous kinds of CSCs, not only breast [169,170] but also pancreatic, liver [171], osteosarcoma [172], ovarian [173], and glioblastoma [174] cancer stem cells, in which its failure negatively influences the expression of staminal markers and, thus, the ability of self-renewal. In hematological cancers, autophagy can act as a tumor suppressor and in chemoresistance. 

Recently, researchers have focused on dissecting the underlying mechanisms of autophagy-dependent CSC maintenance, and various pathways have been identified. For instance, Yeo and collaborators [177] showed that autophagy performs via EGFR/Stat3 and Tgfβ/Smad signaling in two breast cancer stem-like cells (ALDH^+^and CD29hiCD61^+^, respectively). The authors demonstrated reduced phosphorylation of EGFR depleting *FIP200*, which, in turn, led to reduced STAT3 activation and, consequently, altered ALDH^+^breast CSCs tumorigenicity. Autophagy inhibition induces reduced TFGβ2 and TGFβ3 expression, causing a defect in Smad signaling, which is crucial for the CD29hiCD61+CSC phenotype. Furthermore, in the triple-negative breast CSCs, it has been found that by inhibiting autophagy, the secretion of IL-6 was reduced, probably via the STAT3/JAK2 pathway [178]. The secretion of this cytokine is important for CSC survival [179] and necessary to induce the CD44^+^/CD24 low phenotype in breast cancer cell lines, thus confirming that the IL-6-JAK2-STAT3 pathway may have a key role in the conversion of non-CSCs into CSCs.

It has been suggested that the role of FOXO proteins in autophagy and, in particular, the correlation among FOXO, autophagy, and cancer is emerging (extensively reviewed in [180,181,182]). In particular, FOXO proteins controlling autophagy regulate cancer growth and metastasis [180,181,182]. In other investigations, a key role for FOXOs in controlling the fate of CSCs has been indicated [183]. FOXO-dependent regulation of transcription is essential to maintain the homeostasis of stem cells in both embryos and adults [184]; however, the role of FOXO in affecting CSC functions needs to be clarified. The knockdown of FOXO3 led to sustained CSC self-renewal in glioblastoma, prostate, liver, and colorectal cancers and female cancers [185,186,187,188]; in contrast, leukemia-initiating cells require FOXO3 for stem cell survival [189,190]. However, further investigations are required to comprehend how FOXO-dependent regulation of stemness and autophagy mechanisms are interrelated in tumorigenesis. Interestingly, a correlation between autophagy, stemness markers, and NAD^+^biosynthesis has been reported. Sharif and his group revealed that altering basal autophagy by inhibitors or activators, the pluripotency of teratocarcinoma CSCs was strongly reduced, inducing a state of differentiation or/and senescence [191]. 

Another research group studying ovarian CSCs has reported a correlation between autophagy and stemness [173]. In particular, Forkhead Box A2 (FOXA2) is overexpressed in ovarian CSCs and controlled by autophagy; indeed, the inhibition of autophagy by pharmacological and genetics methods led to FOXA2 reduction and, in turn, loss of self-renewal capability. Finally, other studies suggested a role for autophagy in controlling chromosome stability; thus, CSCs may activate autophagy to avoid additional DNA damage and, thus, preserve their maintenance [192].

### 5.2. Mitophagy

As mentioned above, autophagy performs a dual role in cancer as tumor suppressor and promoter. Indeed, depending on the context, it may positively or negatively influence tumor growth and invasion. CSCs are usually characterized by a deregulation of pathway autophagy/mitophagy [193]. The modulation of this pathway controls CSC generation, differentiation, plasticity, migration/invasion, and immune and drug resistance. Mitophagy is crucial in the control of normal tissue stem cell homeostasis. Mitophagy controls mitochondria functionality and cellular metabolism [168]. For instance, removing altered mitochondria (the main source of ROS) by mitophagy inhibits senescence and regulates ROS-induced genome injury [194]. Avoiding ROS-induced injuries is essential for the preservation of stemness. A crucial role for mitophagy was reported during the glycolytic switch necessary for mouse neurogenesis [195]. The turnover of mitochondria through mitophagy supports the maintenance of stemness by limiting the capacity of the stem cells for oxidative phosphorylation, increasing the glycolytic activity for energy requests. Inhibiting the mitophagy process, CD44 expression was reduced and the translocation of p53 to the nucleus was supported, where it antagonized the expression of stemness markers [168].

## 6. CSC-Targeting Therapeutic Approaches 

As mentioned above, several current therapies fail to eradicate tumors due to the ability of CSCs to escape different programmed cell deaths. Thus, developing CSC-targeting therapeutic approaches appears of primary importance. In the following paragraphs, we report encouraging studies of CSC-targeting that use both natural and synthetic compounds (Table 2). 

### 6.1. Natural Compounds 

Different nutraceuticals are efficient as adjuvant agents for cancer therapy in in-vivo and in-vitro studies [230,231,232,233]. A diet abundant in vegetables and fruits is related to cancer risk reduction; however, the mechanisms are still unclear. In particular, sera of transgenic adult mice with a diet rich in soy isoflavone genistein or blueberry polyphenol showed an altered population of stem-like/progenitor cells [234]. In addition, these natural compounds were able to reduce the spheroid formation of breast CSCs [234,235]. 20(S)-ginsenoside Rg3, one of the major components extracted from *Panax ginseng*, has been shown to have anticancer effects in colon CSCs and induce apoptosis by controlling numerous signaling pathways, including caspases-9 and -3 and P53 [196]. Ginsenoside Rb1 and its metabolite compound K inhibited ovarian CSC self-renewal and exposed the CSCs to high doses of chemotherapeutics. The effect of ginsenoside Rb1 on CSCs is related to the inhibition of the Wnt/β-catenin signaling pathway by reducing β-catenin/T-cell factor-dependent transcription and the expression of its target genes (ATP-binding cassette G2 and P-gp) [198]. More recently, ginsenoside Rg3 has been demonstrated to inhibit the growth and stemness of colorectal CSCs both in vivo and in vitro [197].

Notably, epigallocatechin-3-gallate (EGCG), the main component of green tea, reduced lung CSC spheroid size, decreased stemness markers, inhibited proliferation, and stimulated apoptosis. The mechanism is mainly ascribed to the alteration of the Wnt/β-catenin pathway [199]. EGCG also suppressed the self-renewal ability of neck and head CSCs, reducing the expression of stemness markers and controlling spheroid formation. Furthermore, EGCG enhanced cisplatin-induced chemosensitivity by inhibiting ABC transporter genes and reducing tumor growth, and it activated apoptosis in xenograft models. The authors have ascribed the main mechanism of the anti-CSC activity of EGCG to the suppression of the Notch pathway [200].

Resveratrol is a polyphenol widely used in the traditional Mediterranean diet, which has shown different positive effects [236]. Resveratrol inhibited the Wnt/β-catenin signaling pathway in mammospheres in vitro and in vivo and, thus, induced autophagy [201]. Moreover, this polyphenol reduced glioma CSC proliferation and motility by controlling Wnt signaling and EMT activators in glioblastoma multiforme lines [202]. Furthermore, the efficacy of resveratrol mixed with grape seed extract was studied in isolated human colon CSCs in vitro and in a mouse model of colon carcinogenesis; interestingly, this combination suppressed Wnt/β-catenin and led to the mitochondrial-mediated apoptosis of CSCs [203]. Broussoflavonol B, a chemical extracted from the *Broussonetia papyrifera* tree, reduced estrogen receptor (ER)-α36 expression and counteracted ER-negative breast cancer stem-like cell growth, inducing apoptosis [204]. Curcumin, extracted from turmeric, induced CD133+ rectal CSC apoptosis and significantly increased radiosensitivity of rectal CSCs [205]. Curcumin also sensitized breast CSCs to chemotherapy, reducing ABCG2 expression [206]. 

Furthermore, the flavonoid morusin stimulates apoptosis of cervical CSCs, increasing caspase-3 and Bax in a dose-dependent manner and reducing NF-κB/p65 and Bcl-2 [207]. In addition, morusin was able to constrain human glioblastoma CSC growth in vitro and in vivo, inhibiting stemness markers and adipocyte differentiation and inducing apoptotic death [208]. Upon morusin treatment, CSCs showed increased activity of Bax and caspase-3 in parallel with a decreased activity of Bcl-2 [208].

Diallyl trisulfide (DATS), a garlic-derived organosulfur, showed anticancer activities. Indeed, DATS decreased spheroid size, inhibited proliferation, diminished CSC markers, and triggered apoptosis by inhibiting the Wnt/β-catenin pathway and its target genes in colorectal CSCs [209]. Comparably, DATS reduced viability and proliferation, diminished CSC marker expression, and stimulated apoptosis in human breast CSCs by inhibiting the Wnt/β-catenin pathway [210].

### 6.2. Synthetic Inhibitors

IMD-0354, an inhibitor of IKKB, modulating IKKB and NF-κB, leads to breast CSC apoptosis [211]. LDE225 (also named Erismodegib or NVP-LDE-225) is a new specific Hedgehog signaling pathway inhibitor and Smoothened antagonist. LDE225 reduces EMT and human prostate CSC growth and spheroid formation in NOD/SCID IL2Rγ null mice by controlling pro- and antiapoptotic proteins [212]. 

NV-128 is a synthetic flavonoid derivative that targets mitochondria in CD44^+^/MyD88^+^ ovarian CSCs and stimulates apoptosis by supporting cellular cell starvation, which, in turn, triggers two independent pathways: the mitochondrial MAP/ERK pathway, inducing the loss of mitochondrial membrane potential, and the AMPKα1 pathway, inducing mTOR suppression [213]. 

Disulfiram (DS) is an orally bioavailable ALDH inhibitor that is a thiocarbamate alcoholism drug [214]. It has been reported that DS inhibits the P-gp extrusion pump, blocks NF-κB, sensitizes to chemotherapy, decreases angiogenesis, and reduces tumor growth in mice [215]. DS is able to inhibit both ALDH2 and ALDH1 isozymes, which are upregulated in CSCs, suggesting the potential use of DS as an antineoplastic drug [237]. Numerous papers have reported DS anti-cancer effects in different tumors (widely reviewed in [216,217]). The antineoplastic effects of DS are mainly due to the induction of high intracellular ROS levels, thus leading the CSCs towards apoptosis [218]. It has been demonstrated that a DS/copper complex can target ALDH1A1 and reduce tumor relapses that are mainly led by ALDH-high CSCs [238]. In ovarian cancers, DS showed cytotoxic effects similar to chemotherapeutics (i.e., cisplatin, paclitaxel) and also showed a target effect on cancer cells without affecting normal cells [239]. The cytotoxic effect of DS is mainly due to programmed cell death activation; an additive impact in combination with chemotherapy was detected [240]. For instance, DS used in combination with chemotherapeutic 5-fluorouracil (5-FU) showed an increased apoptotic effect on human colorectal cancer cell lines (DLD-1 and RKO (WT) cells) and increased the cytotoxicity of 5-FU. DS was also able to reduce 5-FU chemoresistance in a chemoresistant cell line H630(5-FU) [215]. In vitro treatment with DS/copper substantially reduced the expression of stem cell markers (Sox2, Oct-4, and Nanog) and lowered the ability of nonsmall cell lung cancer (NSCLC) stem cells for proliferation, invasion, and self-renewal. In NOD/SCID xenograft models of NCI-H1299 cells, DS/copper was administered, and, interestingly, it removed ALDH-positive cells, decreased tumor growth, and abolished tumor recurrence [241].

Liu and collaborators analyzed the cytotoxic effect of DS/copper and DS and gemcitabine on GBM stem-like cells. DS/copper boosts the cytotoxicity of gemcitabine. Combination index–isobologram analyses have suggested a synergistic effect between DS/copper and gemcitabine. The authors showed that the cytotoxicity effects of the combined drugs led to increased apoptosis in GBM stem-like cells, which may be due to increased ROS and downregulation of both ALDH and the NFκB pathway. These data indicate that DS/copper may induce the intrinsic apoptotic pathway through modulation of the Bcl2 family, and it is also able to eliminate the stem-like cell population in GBM cell lines [242]. Another group reported the potentiated effect of DS in combination with copper. In particular, Hothi and collaborators evaluated its effect on glioma stem cells and showed increased apoptosis against CSCs specifically [219]. In addition, DS/copper reduced the chymotrypsin-like proteasomal activity in these cells, parallel with the inhibition of the ubiquitin–proteasome pathway and the consequent tumor cell death [219]. 

Another investigation indicated that DS/copper treatment in mammospheres strongly reduced the CSC population. DS in combination with copper generated ROS and triggered the downstream p38 MAPK and apoptosis-related cJun N-terminal kinase pathways. Furthermore, constitutive NFκB activity in breast CSCs was blocked by DS/copper [243].

DS abolished CSC features and totally reversed paclitaxel and cisplatin resistance in MDA-MB-231PAC10 cells (highly cross-resistant to paclitaxel and cisplatin) [244].

PF-03084014, a γ-secretase inhibitor, was able to inhibit the expression of survivin (IAP family protein) and MCL1 (antiapoptotic molecule) and reduce CSCs in triple-negative breast cancer models [220]. Sorafenib and FH535 (β-catenin/Tcf inhibitor) impede liver CSC proliferation and growth by controlling survivin [221]. Similarly, STX-0119, an inhibitor of signal transducer and activator of transcription (STAT) 3, prevents the expression of STAT3 target genes, including survivin and c-Myc, leading to the apoptotic cell death of CSCs (derived from recurrent glioblastoma) [222].

At high doses, 4-diethylaminobenzaldehyde (DEAB), an aldehyde dehydrogenase inhibitor, can deplete CD133+ ovarian CSCs [157]. DEAB was able to significantly reduce spheroid size and proliferation of patient-derived endometrial CSCs [245]. Recently, it has been demonstrated that DEAB eradicates human pancreatic cancer stem-like cells, reducing cell viability and increasing cell apoptosis [223]. In colon cancer, a survivin inhibitor increased the sensitivity in the CD133 + cell segment to 5-fluorouracil [246]. 

Another synthetic inhibitor widely used in CSCs is the all-trans retinoic acid (ATRA), an active metabolite of vitamin A. It is implicated in numerous pivotal pathways such as the immune system, embryonic development, function, epithelial integrity, and reproduction [247]. It has been reported that ATRA constrains tumor cell growth by shifting cell cycle progression [224] and targets CSC spheroids both in vitro and in vivo [225]. ATRA decreased ALDH1 expression and inhibited in-vitro ovarian CSC spheroid size and cell invasion and migration; it also inhibited the tumorigenesis of ovarian cancers in vivo [225]. The authors indicate that the effects are due to the ability of ATRA to downregulate ALDH1/FoxM1/Notch1 signaling [225]. ATRA used in combination with gefitinib in cancer stem-cell-like adenocarcinoma decreased CSC-mediated resistance by reducing ALDH1A1 and CD44 expression and increasing the anticancer effects of gefitinib in NSCLC/ADC [248].

As mentioned above, autophagy is an adaptive mechanism necessary for CSC maintenance. Notably, CQ, an autophagy inhibitor, targeted pancreatic CSCs by inhibiting CXCR4 and autophagy by hedgehog signaling [226]. Salinomycin and metformin were also able to target CSCs in vitro and in vivo and to potentiate the effect of chemotherapeutics in inhibiting tumor growth and prolonging remission [249,250,251]. PTC-209, a BMI-1 inhibitor, prevented colorectal CSC self-renewal and inhibited tumor growth and progression in a mouse xenograft model [227]. 

Regarding necroptosis, different therapeutic approaches targeting this pathway have been reported. For instance, necroptosis was induced in breast CSCs by a Ni(II) dithiocarbamate phenanthroline complex [228]. Similarly, it has been reported that an Os(II) bathophenanthroline complex induced human breast CSC death, predominantly by necroptosis. This drug was able to selectively target mammospheres, reducing CSC proliferation and viability [229]. Notably, Chefetz and collaborators reported that a selective, pan-ALDH1A family inhibitor, 673A (ALDH1Ai), activated necroptosis in CD133^+^ ovarian CSCs [157]. Necroptosis is due, in part, to the induction of mitochondrial uncoupling proteins and oxidative phosphorylation decrease. ALDH1Ai 637 is extremely synergistic with chemotherapy, removing chemotherapy-resistant tumors, counteracting tumor initiation and growth, and improving tumor eradication rates in patients [157]. These studies reinforced the role of ALDH1 family enzymes in chemoresistance and reinforced the hypothesis that therapeutic approaches targeting the ALDH family may ameliorate clinical outcomes in cancer patients.

## 7. Discussion and Conclusions

The evasion of programmed cell death represents a hallmark of human cancers. CSCs are the main character of this process since they are able to evade programmed cell death to promote their self-renewal, supporting survival in response to radio and chemotherapy. Basing on the uncovered evidence, a combination of modulators of programmed cell death and standard chemotherapeutic agents may be an efficient anticancer therapeutic approach. These combined therapies may contribute to counteracting chemotherapy resistance and sensitize CSCs to anticancer treatments. However, there are several points that need to be deeply investigated in order to develop an effective anticancer treatment targeting programmed cell death; in particular, it is necessary to consider the tumor microenvironment, cell type, stimuli and/or stress conditions, and tumor growth properties. Controlling the programmed cell death machinery (including apoptosis, autophagy, and necroptosis pathways) to eradicate CSCs has shown great potential. The suppression of their self-renewal, the selective induction of CSC differentiation, or the inhibition of CSC survival and growth by targeting microenvironment elements or key signaling molecules represent undoubtedly attractive investigations to cancer researchers. The selective eradication of cancer cells and, in particular, CSCs can be achieved with both synthetic and natural agents, used as single approaches or in combination with gold-standard chemotherapies that interfere with the deregulated cellular signals that promote CSCs. Additionally, targeted therapies reactivating cell death signaling in CSCs may show synergic effects with conventional therapies and enhance clinical efficacy.

Even if the reported studies are encouraging, further investigations are necessary to establish a combination of agents that is able to eradicate CSCs or inhibit their growth and proliferation.

## Figures and Tables

**Figure 1 cancers-13-00328-f001:**
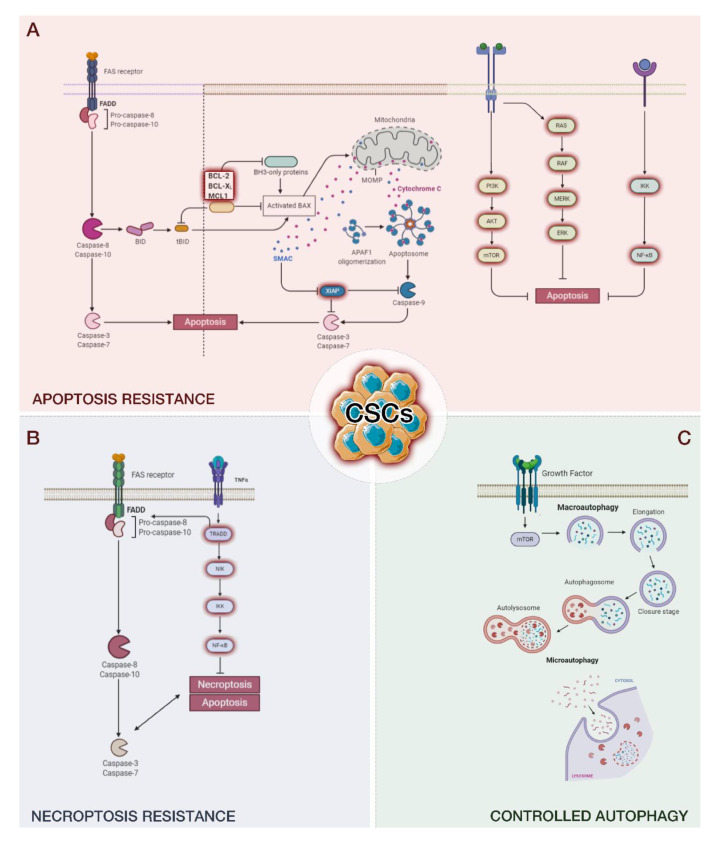
The ability of cancer stem cells (CSCs) to escape different programmed cell deaths, including apoptosis (**A**), necroptosis (**B**), and autophagy (**C**). The remaining CSCs promote cancer recurrence.

**Table 1 cancers-13-00328-t001:** Types of cancer in which CSCs have been detected and the main markers.

Types of Cancer	Markers	References
Leukemia	CD34, CD38, CD123, TIM3, CD25, CD32, and CD96	[15,16,17]
Breast	CD44, CD24, and ALDH1A1	[18,19,20,21]
Prostate	PSA^-/lo^, integrin α2β1, and CD133	[22,23]
Melanoma	ABCB5, Nanog, Oct3/4, and CD133	[24,25]
Brain	CD90, CD44, and CD15	[26,27,28,29]
Lung	CD44, CD133, and CD90	[30]
Liver	EpCAM, E-cadherin, CD133, CD29, and CK	[31,32,33]
Pancreas	CD44, CD24 and epidermal surface antigen, CD133, and CXCR4	[34,35,36]
Kidney	Nestin, CD133, CD24a, CXCR7, CD44, and Pax2	[37,38]
Ovary	CD133, CD44, CD117, EpCAM, LGR5, ALDH1/2, LY6A, and CD24	[39,40]

**Table 2 cancers-13-00328-t002:** CSC-targeting therapeutic approaches.

Compound	Origin	Target/Effects	References
20(S)-ginsenoside Rg3	*Panax ginseng*	Caspases and P53	[196,197]
Ginsenoside Rb1	*Panax ginseng*	Wnt/β-catenin Reduces stemness markers	[198]
Epigallocatechin-3-gallate	Green tea	Wnt/β-catenin Inhibits ABC transporter Reduces stemness markers	[199,200]
Resveratrol	Cocoa, grapes, peanuts, berries	Wnt/β-catenin Induction of autophagy	[201,202,203]
Broussoflavonol B	*Broussonetia papyrifera* tree	Inhibits growth	[204]
Curcumin	*Curcuma longa*	ABCG2 Increases radio- and chemo-sensitivity of CSCs	[205,206]
Morusin	*Ramulus mori*	Increases caspase-3 and BaxDecreases Bcl-2 and NF-κB/p65 Inhibits stemness markers	[207,208]
Diallyl trisulfide	*Allium sativum*	Wnt/β-catenin Inhibits stemness markers	[209,210]
IMD-0354	Synthetic	IKKB and NF-κB	[211]
LDE225	Synthetic	Bmi-1	[212]
NV-128	Synthetic	MAP/ERK, AMPKα1, and mTOR	[213]
Disulfiram	Synthetic	P-gp, NF-κB, ALDH, p38, and cJun Inhibits stemness markers Inhibits ubiquitin–proteasome pathway	[214,215,216,217,218,219]
PF-03084014	Synthetic	Survivin, MCL1	[220]
Sorafenib and FH535	Synthetic	Survivin	[130,221]
STX-0119	Synthetic	STAT3, survivin, and c-Myc	[222]
4 diethylaminobenzaldehyde	Synthetic	Bcl-2 and Bax	[223]
All-trans retinoic acid	Synthetic	ALDH1/FoxM1/Notch1	[224,225]
Chloroquine	Synthetic	Autophagy (Hedgehog pathway)	[226]
PTC-209	Synthetic	BMI-1	[227]
Ni(II) dithiocarbamate phenanthroline	Synthetic	Necroptosis	[228]
Os(II) bathophenanthroline	Synthetic	Necroptosis	[229]
673A		Necroptosis, ALDH1	[157]

Abbreviations: ABCG2, ATP-binding cassette transporter G2; IKKB, inhibitor of nuclear factor kappa-B kinase; NF-Κb, nuclear factor kappa-light-chain-enhancer of activated B-cells; P-gp, P-glycoprotein; Bcl-2, B-cell lymphoma 2; Bax, B-Cell lymphoma-associated X; MCL-1, myeloid cell leukemia 1; STAT3, signal transducer and activator of transcription 3; ALDH, aldehyde dehydrogenase.

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
