# Peer review of "The Great Escape: The Power of Cancer Stem Cells to Evade Programmed Cell Death"

_cancers, 2021, doi:10.3390/cancers13020328_

Round 1

Reviewer 1 Report

The review by Castelli et al. describes how cancer stem cells may evade the programmed cell death. The paper is solid and reads well. However, there are several concerns that should be clarified.  

Major concerns:

  1. In the introduction, authors should provide a brief historical overview on the identification of cancer stem cells (tumor initiating cells (TICs)).
  2. In the introduction, the authors should clearly state the aim of their study.
  3. 3. subsection. The omission of melanoma cells and ABCB5 is completely incomprehensible.
  4. The authors should consider adding a table summarizing for which cancers, cancer stem cells have been confirmed.
  5. Table 1. Is anything known about the clinical trials with the compounds shown in a Table 1?
  6. Discussion and Conclusion section is far too short and superficial. Authors should expand it, and write about the challenges the clinicians are facing in the targeting of cancer stem cells etc.

Minor concerns:

  1. Lines 92-93 “In particular, CSCs show an overexpression (…)” Please specify mechanisms that lead to overexpression (transcription rate, mulitplicatation of genes etc.).
  2. Lines 124-125 “However, in some tumours it has been (…). Please specify these tomours.

Author Response

Reviewer 1

The review by Castelli et al. describes how cancer stem cells may evade the programmed cell death. The paper is solid and reads well. However, there are several concerns that should be clarified.  

Response: We thank the Reviewer for the time spent in revising our review article and for the valuable comments provided that helped us in improving our manuscript. We tried to address all the points raised.

Major concerns:

  1. In the introduction, authors should provide a brief historical overview on the identification of cancer stem cells (tumor initiating cells (TICs)).

Response: Thank you for the comment. We now provided a brief paragraph regarding the identification of cancer stem cells as suggested.

  1. In the introduction, the authors should clearly state the aim of their study.

Response: Thank you for the comment. We now specified the aim of our review article.

  1. 3. subsection. The omission of melanoma cells and ABCB5 is completely incomprehensible.

Response: We apologize for the oversight. We now added this information in the manuscript.

  1. The authors should consider adding a table summarizing for which cancers, cancer stem cells have been confirmed.

Response: We appreciate the Reviewer’s suggestion, and we added a Table as suggested.

  1. Table 1. Is anything known about the clinical trials with the compounds shown in a Table 1?

Response: To our knowledge there are no clinical trials targeting specifically CSCs.

  1. Discussion and Conclusion section is far too short and superficial. Authors should expand it, and write about the challenges the clinicians are facing in the targeting of cancer stem cells etc.

Response: Thank you for the comment. We now expanded the Discussion and Conclusion section as suggested.

Minor concerns:

  1. Lines 92-93 “In particular, CSCs show an overexpression (…)” Please specify mechanisms that lead to overexpression (transcription rate, mulitplicatation of genes etc.).

Response: Thank you for the comment. We now provided this information.

  1. Lines 124-125 “However, in some tumours it has been (…). Please specify these tomours.

Response: Thank you for the comment. We now specified the kind of tumours.

Reviewer 2 Report

In their manuscript entitled « The great escape: the power of cancer stem cells to evading cell death », the authors aim at reviewing the recent literature regarding CSC and cell death resistance. They first introduce a general definition of CSCs and their characteristics and insist on the importance of the cross-talk between CSCs and their microenvironment. They then describe in detail means by which CSC resist to the induction of apoptosis, necroptosis and autophagy-associated cell death. The authors then describe several molecular targets and pharmacological means, including natural compounds, to eradicate CSCs. This review is overall nicely written and comprehensive. I do have some comments detailed below which I hope could help the authors to further improve the depth of this review. A couple of typos also need to be corrected.

As a general comment, one regulated cell death pathway lacking here and that is involved in controlling CSCs is ferroptosis. A paragraph should be added about it.

Line 56 the authors state »the self-renewal and clonogenicity properties are due to the microenvironment ». If this was the case, isolation of CSC and evaluation of such properties in vitro would be impossible. I think this sentence needs to be rephrased.

Line 76: Figure 1 is barely readable. The authors may want to split this figure in three different figures depicting the different types of cell death signalling, which could then be further precisely detailed,  and place these at relevant places for each part of their manuscript.

Line 89: Is the transformation of normal stem cell the only way to create apoptosis resistant CSC? There is an increasing amount of papers showing that differentiated tumour cells can actually dedifferentiate to become CSCs. The authors may want to comment on this aspect in their review: this plasticity is likely to be accompanied by an increased ability to resist cell death (and so the signalling pathway activated to drive such a dedifferentiation process might trigger this as well).

Line 112 “and homo”, read like extra words, maybe Omi is meant here?

Line 124: caspase-independent process rather than independent caspase process. One/several reference(s) should be added here. The authors may also want to consider instances in which MOMP actually does not trigger death (could such mechanisms/occurrences be enhanced in CSCs? This could be discussed) For a recent example: Berthenet K et al, Cell Reports, 2020 and several other older studies by Stephen Tait’s team.

Line 133: The DISC does not necessarily comprise TRADD for TNFR1 signalling and is actually a secondarily formed cytosolic complex deprived of TNFR1. Multiple more recent reviews could be used here for precise description of DR signalling (e.g Lafont E, Cancers, 2020; Ting A and Bertrand MJM, Trends Immunol, 2016, Annibaldi A and Walczak H, Cold Spring Harbor Perpectives, 2020).

Line 146: “in CSCs” is redundant and should be removed.

Line 154:  Given to the role -> Given the role.

Line 157: to be involved -> involved

Line 175 redux -> redox

Line 185: this sentence is not clear and should be rephrased.

Line 204: XIAP does not necessarily have the strongest anti-apoptotic activity. It depends on the stimuli (in the context of TNF signalling, depletion of cIAP1/2 can be a very potent apoptosis driver). The authors may want to specify that it instead has the most direct anti-apoptotic activity instead.

Line 236: necroptosis is not a “combination of apoptosis and necroptosis”, but a regulated form of necrosis.

Line 248 involving -> involved

Line 272 promote -> promoted

Line 283 maintain -> maintained

Paragraph from line 316: The link between this part and autophagy is not clear, the authors should specify it, otherwise, I am not sure it is meaningful to include these studies on FOXOs.

Author Response

Reviewer 2

In their manuscript entitled « The great escape: the power of cancer stem cells to evading cell death », the authors aim at reviewing the recent literature regarding CSC and cell death resistance. They first introduce a general definition of CSCs and their characteristics and insist on the importance of the cross-talk between CSCs and their microenvironment. They then describe in detail means by which CSC resist to the induction of apoptosis, necroptosis and autophagy-associated cell death. The authors then describe several molecular targets and pharmacological means, including natural compounds, to eradicate CSCs. This review is overall nicely written and comprehensive. I do have some comments detailed below which I hope could help the authors to further improve the depth of this review. A couple of typos also need to be corrected.

Response: We thank the Reviewer for the time spent in revising our review article and for the valuable comments provided that helped us in improving our manuscript. We tried to address all the points raised.

As a general comment, one regulated cell death pathway lacking here and that is involved in controlling CSCs is ferroptosis. A paragraph should be added about it.

Response: We thank the reviewer for the comment and we totally agree. We now added a paragraph regarding CSCs and ferroptosis as suggested.

Line 56 the authors state »the self-renewal and clonogenicity properties are due to the microenvironment ». If this was the case, isolation of CSC and evaluation of such properties in vitro would be impossible. I think this sentence needs to be rephrased.

Response: Thank you for the comment. We now rephrased the sentence as suggested.

Line 76: Figure 1 is barely readable. The authors may want to split this figure in three different figures depicting the different types of cell death signalling, which could then be further precisely detailed,  and place these at relevant places for each part of their manuscript.

Response: Thank you for the comment. We now rearranged the figure to make it more readable and we add a letter to recall them in the manuscript.

Line 89: Is the transformation of normal stem cell the only way to create apoptosis resistant CSC? There is an increasing amount of papers showing that differentiated tumour cells can actually dedifferentiate to become CSCs. The authors may want to comment on this aspect in their review: this plasticity is likely to be accompanied by an increased ability to resist cell death (and so the signalling pathway activated to drive such a dedifferentiation process might trigger this as well).

Response: We appreciate the Reviewer’s comment and totally agree. We now added this aspect in the review as suggested.

Line 112 “and homo”, read like extra words, maybe Omi is meant here?

Response: We apologize for the oversight and we now corrected in the manuscript.

Line 124: caspase-independent process rather than independent caspase process. One/several reference(s) should be added here. The authors may also want to consider instances in which MOMP actually does not trigger death (could such mechanisms/occurrences be enhanced in CSCs? This could be discussed) For a recent example: Berthenet K et al, Cell Reports, 2020 and several other older studies by Stephen Tait’s team.

Response: Thank you for the suggestion. We added the references as suggested and we have also added a brief discussion regarding failed apoptosis occurring in cancer as suggested.

Line 133: The DISC does not necessarily comprise TRADD for TNFR1 signalling and is actually a secondarily formed cytosolic complex deprived of TNFR1. Multiple more recent reviews could be used here for precise description of DR signalling (e.g Lafont E, Cancers, 2020; Ting A and Bertrand MJM, Trends Immunol, 2016, Annibaldi A and Walczak H, Cold Spring Harbor Perpectives, 2020).

Response: We thank the Reviewer for the suggestion. We now added more information regarding DISC DR signalling as suggested.

Line 146: “in CSCs” is redundant and should be removed.

Response: Thank you for the suggestion. We now removed the redundant term.

Line 154:  Given to the role -> Given the role.

Response: Thank you for the comment. We now changed accordingly.

Line 157: to be involved -> involved

Response: Thank you, we modified accordingly.

Line 175 redux -> redox

Response: Thank you, we now modified accordingly.

Line 185: this sentence is not clear and should be rephrased.

Response: We apologize, and we now rephrased as suggested.

Line 204: XIAP does not necessarily have the strongest anti-apoptotic activity. It depends on the stimuli (in the context of TNF signalling, depletion of cIAP1/2 can be a very potent apoptosis driver). The authors may want to specify that it instead has the most direct anti-apoptotic activity instead.

Response: Thank you for the comment. We now modified the paragraph accordingly.

Line 236: necroptosis is not a “combination of apoptosis and necroptosis”, but a regulated form of necrosis.

Response: Thank you for the comment. We now modified in the manuscript.

Line 248 involving -> involved

Response: Thank you for the comment. We modified the word in the manuscript.

Line 272 promote -> promoted

Response: Thank you for the comment. We modified the word in the manuscript.

Line 283 maintain -> maintained

Response: Thank you for the comment. We corrected the error accordingly.

Paragraph from line 316: The link between this part and autophagy is not clear, the authors should specify it, otherwise, I am not sure it is meaningful to include these studies on FOXOs.

Response: We now specified the link as suggested. We hope now it is clear why we included this section in the paragraph in object.

Reviewer 3 Report

  1. This manuscript is interesting and well-done.
  2. Theme of this research, to suggest a guideline of the resistance against anti-cancer drugs on cancer stem cells.
  3. In this study, authors explain that several current therapies fail to eradicate tumors due to the cancer stem cells 11 (CSCs) ability to escape different programmed cell death.
  4. The reason why the current chemotherapy fails is because cancer stem cells acquire a mechanism to avoid ‘death’ signaling pathway. Drug-resistant cancer stem cells evade chemotherapeutic drug effects via the induction of escape to ‘death’ signaling pathway.
  5. Line 97-98, 146, 186-188, It's just my opinion. If there is a known paper, these sentences need references.
  6. Line 149, ‘P-glycoprotein (P-gp, MDR1, ABCB1)’, is there any reference? If you have a reference, you'd better refer to it.

Author Response

Reviewer 3

  • This manuscript is interesting and well-done.
  • Theme of this research, to suggest a guideline of the resistance against anti-cancer drugs on cancer stem cells.
  • In this study, authors explain that several current therapies fail to eradicate tumors due to the cancer stem cells 11 (CSCs) ability to escape different programmed cell death.
  • The reason why the current chemotherapy fails is because cancer stem cells acquire a mechanism to avoid ‘death’ signaling pathway. Drug-resistant cancer stem cells evade chemotherapeutic drug effects viathe induction of escape to ‘death’ signaling pathway.

Response: We thank the Reviewer for the time spent in revising our review article and for the valuable comments provided that helped us in improving our manuscript. We tried to address all the points raised.

  • Line 97-98, 146, 186-188, It's just my opinion. If there is a known paper, these sentences need references.

Response: Thank you for the comment. We now added references as suggested.

  • Line 149, ‘P-glycoprotein (P-gp, MDR1, ABCB1)’, is there any reference? If you have a reference, you'd better refer to it.

Response: We thank the Reviewer for the comment and we now added references accordingly.

Round 2

Reviewer 1 Report

The authors greatly improved their manuscript, thus I have no further comments.

Reviewer 2 Report

The authors have satisfactorily adressed my comments.